# Bulk Bitwise Accumulation in Commercial DRAM

**Tatsuya Kubo**
The University of Tokyo
tatsuya.kubo@is.s.u-tokyo.ac.jp

**Masayuki Usui**
The University of Tokyo

**Tomoya Nagatani**
The University of Tokyo

**Daichi Tokuda**
The University of Tokyo

**Lei Qu**
Microsoft Research

**Ting Cao**$^*$
Microsoft Research
ting.cao@microsoft.com

**Shinya Takamaeda-Yamazaki**
The University of Tokyo
shinya@is.s.u-tokyo.ac.jp

## Abstract

Processing-in-memory (PIM) is a promising paradigm for addressing data transfer bottlenecks in data-intensive workloads, particularly in machine learning. Among PIM techniques, Processing-using-Commercial-DRAM (PuCD) offers a practical approach for enabling in-memory computing by employing widely available DRAM modules without hardware modifications. With its massive bit-level parallelisms, PuCD has a high-performance capability of bulk bit logic operation. However, implementing *accumulation* operations, crucial for machine learning tasks, remains challenging in PuCD. The need for multiple consecutive operations in accumulation leads to increased latency and error propagation. To address these challenges, we propose a novel method for bulk bitwise accumulation using PuCD. As a fundamental building block for our accumulation method, we introduce a novel implementation of the *population-count-of-3* (POPCNT3) operation tailored for commercial DRAM. On top of this, we present a POPCNT3-based bitwise accumulation method that efficiently handles large input sizes, enabling scalable bitwise accumulation for various input sizes. We evaluate the throughput and errors of our approach using commercial DDR4 DRAM modules with an FPGA. The experiments indicate that the throughput improvement is up to 348 times over A100 GPU across various input sizes with negligible errors to maintain the accuracy of machine learning applications. These results demonstrate that PuCD can provide a practical pathway for accelerating machine learning tasks without requiring specialized memory chips.

## 1 Introduction

Emerging data-intensive workloads, particularly in machine learning (ML), are increasingly constrained by data transfer costs rather than computation. Deep neural networks, especially large language models, exemplify this trend with their demand for processing vast amounts of data, often requiring tens of gigabytes of parameters [Brown et al., 2020]. The predominance of multiply-accumulate (MAC) operations of general matrix-vector multiplication (GeMV) in these models, coupled with their low on-chip data reuse, makes off-chip data transfer the primary performance bottleneck [Choi et al., 2023, Wu et al., 2024].

---

$^*$Corresponding Author

Second Workshop on Machine Learning with New Compute Paradigms at NeurIPS 2024 (MLNCP 2024).

To address this challenge, processing-in-memory (PIM) has reemerged as a promising solution. PIM is a computing paradigm that integrates computational capabilities directly within memory devices, utilizing high internal bandwidth and reducing off-chip data movement. While digital PIM [Devaux, 2019, Kwon et al., 2021, Lee et al., 2022] incorporates processing units within memory devices, analog PIM [Chi et al., 2016, Seshadri et al., 2017, Eckert et al., 2018] transforms memory cells into computational units. Compared to digital PIM, analog PIM offers higher computational density and energy efficiency, while digital PIM provides greater arithmetic precision and flexibility.

While most workloads require high-precision computations, some DNN models, including LLMs, have inherent redundancy that allows them to tolerate lower precision [Ma et al., 2024]. These models can quantize their weights to just a few bits without sacrificing accuracy. This trend towards low-bit arithmetic presents a unique opportunity for analog PIM, which operates with bit-level parallelism, to efficiently accelerate DNNs. Even more intriguing is the discovery of a technology called Processing-using-Commercial-DRAM (PuCD), which transforms standard DRAM into an analog PIM device [Gao et al., 2019, Olgun et al., 2021, Gao et al., 2022, Yuksel et al., 2024, Yüksel et al., 2024]. PuCD enables massive parallelism and substantial computational power by leveraging the density of DRAM, without requiring additional circuitry or changes to the cost-optimized, low-margin DRAM design. By making use of the widespread availability of DRAM and enabling in-memory computation without hardware modifications, PuCD provides a practical solution to bring PIM capabilities into mainstream computing systems.

Despite the potential of PuCD, implementing *accumulation* using PuCD presents significant challenges. Accumulation is a fundamental operation in MAC computations, which form the backbone of many ML processes. Unlike multiplication, which typically involves two inputs, accumulation requires a number of inputs proportional to the matrix size to sum up the multiple results. This characteristic poses two main difficulties for PuCD implementation: performance issues and accuracy concerns. Executing accumulation with many inputs necessitates a proportional number of logical operations, resulting in slow performance when implemented in PuCD. Additionally, PuCD inherently carries computational errors, which can accumulate significantly when multiple operations are performed consecutively, as required in accumulation.

To address the challenges of implementing fast and accurate accumulation operations in PuCD, we propose a novel method for bulk bitwise accumulation. Our approach consists of two key components: First, we introduce an optimized implementation of the *population-count-of-3* (POPCNT3) operation specifically designed for commercial DRAM. POPCNT3 serves as a fundamental building block for our accumulation method, efficiently counting the number of '1' bits in three input bits. Our implementation leverages PuCD to perform POPCNT3 with minimal latency and error propagation. Building upon this optimized POPCNT3 operation, we develop a scalable bitwise accumulation technique capable of handling larger input sizes. This method iteratively applies POPCNT3 operations to groups of inputs, progressively computing higher-order bit positions. By leveraging the efficiency of our POPCNT3 implementation and the parallelism inherent in PuCD, we achieve high-throughput accumulation across various input sizes.

The key contributions of this work are as follows:

1. As a fundamental building block for our accumulation method, we introduce a new implementation of the POPCNT3 operation tailored for commercial DRAM.

2. We present a POPCNT3-based bitwise accumulation method that efficiently handles large input sizes, enabling scalable bitwise accumulation for various input sizes.

3. Using DDR4 DRAM modules, we demonstrate up to 348 times higher throughput than A100 GPU across various input sizes with negligible errors to maintain the accuracy of machine learning applications.

## 2 DRAM Structure and PuCD Technology

Modern computing systems widely employ dynamic random-access memory (DRAM) as their main memory due to its high density and low cost characteristics. DRAM modules utilize a hierarchical structure to enable efficient data management as shown in Figure 1. At the highest level, channels offer independent data paths. Each channel encompasses multiple ranks, with a rank comprising a collection of DRAM chips. Manufacturers divide each chip into multiple banks, which function as

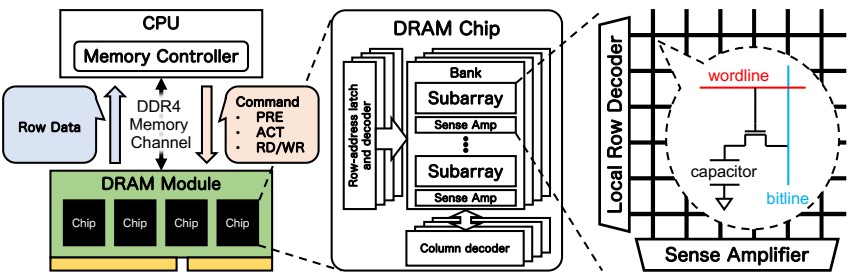

Figure 1: DRAM hierarchy and basic operation. The figure illustrates the structural organization of DRAM from the highest level (memory channel) down to the individual memory cell.

independently operable memory arrays. Banks further subdivide into subarrays, with each subarray consisting of a two-dimensional array of memory cells arranged in rows and columns. The memory cell, the fundamental unit of DRAM, incorporates a single transistor and a capacitor. This cell stores data as the presence or absence of an electrical charge. Each column of memory cells is connected to a bitline, which serves as a communication for reading from and writing to the cells. At the end of each bitline is a sense amplifier, a crucial component that detects and amplifies the small voltage differences on the bitline operations. A memory controller governs data access in DRAM. This controller issues commands in carefully timed sequences. Through this orchestration of commands, the memory controller achieves efficient data access.

Processing-using-Commercial-DRAM (PuCD) is an innovative technique that enables in-memory computation on existing DRAM modules without hardware modifications by exploiting the charge-sharing effect between simultaneously activated rows to perform logical operations. Gao et al. [2019] discovered that logical operations could be performed directly on commercial DDR3 DRAM modules without circuit modifications. Building on this insight, researchers have demonstrated the ability to execute `RowCopy` [Gao et al., 2019, Yuksel et al., 2024], `AND/OR` [Gao et al., 2022], and `NOT` [Yüksel et al., 2024] operations on DDR4 DRAM modules. The charge-sharing effect occurs when multiple DRAM rows are activated simultaneously, causing the electrical charges stored in the capacitors of these rows to redistribute across the bitlines. PuCD implements this charge-sharing effect through simultaneous multi-row activation (SiMRA), a method that involves issuing DRAM commands in a way that violates conventional timing constraints. This technique activates multiple DRAM rows simultaneously on commercial DRAM modules, inducing charge sharing on the bitlines. PuCD then utilizes this charge sharing effect to execute *majority-of-X* (`MAJX`) operations. `MAJX` operations, which determine the majority value among input bits, serve as the basic logical units in PuCD. Prior works build upon `MAJX` to implement fundamental logical operations such as `AND` and `OR`. For instance, they can achieve `AND/OR` operations by fixing certain inputs of `MAJX` to specific values. By combining these basic operations, PuCD can perform more complex computations. However, PuCD faces challenges in operational reliability. The reliability of `MAJX` operations heavily depends on input patterns, with performance degrading significantly when the numbers of '0's and '1's in the input are closely balanced. Furthermore, complex operations often require numerous consecutive `MAJX` operations, potentially leading to cumulative errors and performance deterioration.

## 3 Bulk Bitwise Accumulation

In this section, we introduce a novel approach to perform efficient bulk bitwise accumulation using PuCD. We present our method in two stages: first, we propose an optimized implementation of `POPCNT3`, a fundamental operation that counts the number of '1's in three input bits. Then, we build upon this `POPCNT3` operation to develop a scalable bitwise accumulation technique capable of handling larger input sizes. By combining these two components, we achieve an efficient and accurate method for bulk bitwise accumulation for various input sizes.

### 3.1 `POPCNT3` in Commercial DRAM

We propose a novel method for efficiently implementing `POPCNT3` in PuCD. `POPCNT3` represents a bit operation that counts the number of '1's in three input bits and expresses the result in two bits. `POPCNT3` is defined for three input bits A, B, and C using the following logical expressions:

$$\text{MSB}(A, B, C) = (A \wedge B) \vee (B \wedge C) \vee (C \wedge A)$$

$$\text{LSB}(A, B, C) = (A \wedge \neg B \wedge \neg C) \vee (\neg A \wedge B \wedge \neg C) \vee (\neg A \wedge \neg B \wedge C) \vee (A \wedge B \wedge C)$$

where MSB denotes the most significant bit and LSB the least significant bit. Naive PuCD approaches attempting to implement these logical expressions directly require numerous consecutive `AND`, `OR`, and `NOT` operations. Specifically, LSB calculation necessitates 11 consecutive `AND/OR` operations, leading to high latency and reduced accuracy. In PuCD, multiple consecutive logical operations increase the overall execution time. Moreover, each logical operation carries a success probability. For instance, even if a DRAM module can perform `AND/OR` operations with an exact 99% success rate, the probability of all 11 consecutive operations succeeding drops to approximately 90%. These challenges highlight the inefficiency and accuracy issues of directly implementing `POPCNT3` in PuCD.

To achieve fast and high-precision `POPCNT3` in PuCD, we propose a novel implementation method that leverages `MAJX` operations in PuCD. This approach introduces reference rows and performs `MAJX` operations combining input and reference rows to directly and efficiently calculate the `POPCNT3` output. Figure 2 illustrates the execution timeline of `POPCNT3`. Our method prepares three input rows, two reference rows, and two output rows within the same subarray. The process unfolds as follows: ① We store input values in the input rows. We can move these values can from other rows within the same subarray using `RowCopy`. ② We execute `MAJ3` on the input rows to calculate the MSB value and store the result in the MSB row. ③ We reload the input values into the input rows and copy the `NOT` of the previous MSB value to the reference rows using `NOT`. ④ We perform a `MAJ5` operation combining input and reference rows to calculate the LSB value. Table 1 presents the truth table for `POPCNT3`. The essence of this approach lies in recursively comparing the number of '1's in the input to identify the output bits in binary representation from the most significant bit. We can consider this process as a binary search. Our proposed method executes `POPCNT3` with significantly fewer logical operations (two `MAJX` operations). This reduction in operations substantially decreases the overall execution time. Moreover, by minimizing the number of consecutive PuCD logical operations, we also mitigate output accuracy degradation.

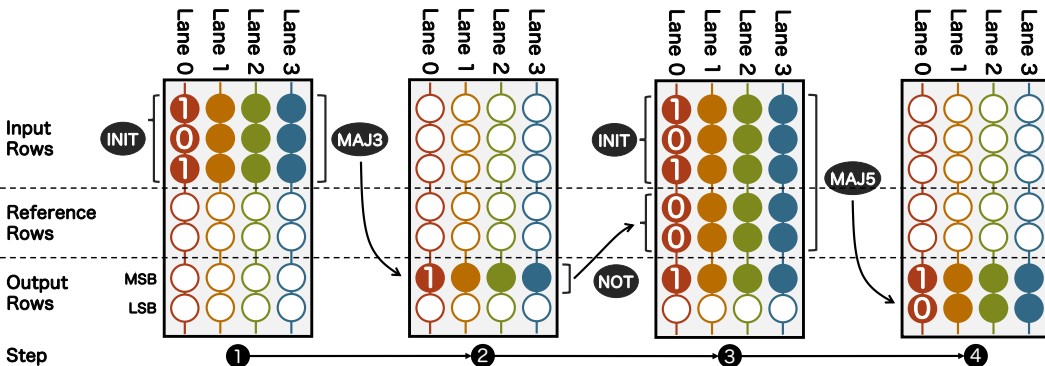

Figure 2: Timeline illustrating data changes in a DRAM segment during a `POPCNT3` operation. The figure shows the step-by-step process of executing a `POPCNT3` operation using PuCD.

### 3.2 `POPCNT3`-based Bitwise Accumulation

Building upon the `POPCNT3` operation, we propose `POPCNT3`-based bitwise accumulation, a method for executing bitwise accumulation on a larger number of inputs. This approach enables counting for a larger number of inputs by iteratively applying `POPCNT3`. While `POPCNT3` generates a 2-bit output from three inputs, we can continue to calculate higher bit position values by repeatedly executing `POPCNT3` on each output bit position. Figure 3 illustrates the procedure for bitwise accumulation

| # of 1s in Input Rows | | 0 | 1 | 2 | 3 |
|---|---|---|---|---|---|
| Reference Row | Value | 1 | 1 | 0 | 0 |
| | | 1 | 1 | 0 | 0 |
| Output Bits | MSB | 0 | 0 | 1 | 1 |
| | LSB | 0 | 1 | 0 | 1 |

Table 1: Truth table for the `POPCNT3` operation in PuCD. This table illustrates the relationship between the number of '1' bits in the input rows, the reference row bit, and the resulting output bit pattern in the `POPCNT3` operation.

with 15 inputs. First, we initially execute `POPCNT3` five times on groups of three inputs from the 15 inputs, yielding five 1st bits and five 2nd bits as outputs. Then, we apply `POPCNT3` to three of the five 1st bits, generating an additional 1st bit and 2nd bit. Finally, we repeat this process, ultimately obtaining a 4-bit output. Consequently, our approach offers an efficient solution for bitwise accumulation across various input sizes, leveraging the simplicity and effectiveness of the `POPCNT3` operation.

Our `POPCNT3`-based bitwise accumulation inherits the bulk processing capabilities and bank parallelism of PuCD, enabling simultaneous computation across all columns in an array and parallel execution across multiple banks. This parallelism allows our method to achieve high throughput for bitwise accumulation operations. However, it's constrained by PuCD's requirement that all inputs and intermediate results must reside within the same subarray. Given that typical DRAM subarrays contain only a few hundred rows, this limits the size of accumulations that can be performed solely using PuCD. For larger accumulations, alternative or hybrid approaches may be necessary, highlighting the importance of careful data layout consideration in practical applications of this technique.

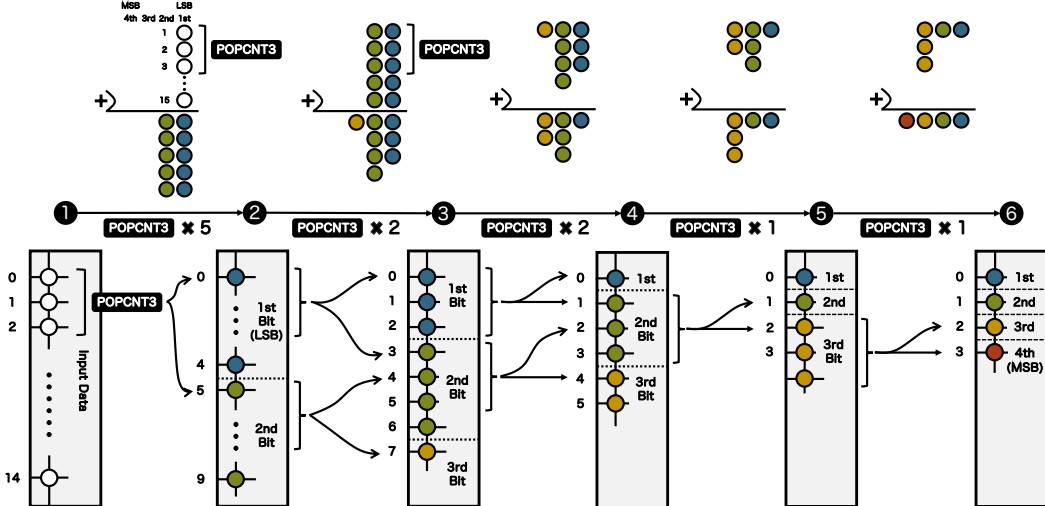

Figure 3: Timeline of a 15-input bitwise accumulation operation in a single DRAM bitline using `POPCNT3`-based method. This figure illustrates the step-by-step process of performing bitwise accumulation on 15 inputs using our proposed `POPCNT3`-based approach.

## 4 Results and Discussion

We evaluate the performance and accuracy of our proposed bulk bitwise accumulation method using DDR4 DRAM for different accumulation sizes. We refer to a single bulk bitwise accumulation computation as a *kernel*. We express the kernel size as $K \times N$, where $K$ represents the input length and $N$ denotes the bulk data width. To implement our proposed method, we use an FPGA connected

to four DDR4 DRAM modules. We compare the performance of our method with an NVIDIA A100 GPU as a baseline. Further details about our evaluation setup are in Appendix A.

Table 2 presents the throughput and latency of our proposed method for each kernel size. Throughput indicates the amount of data that can be processed per second. For our method, we calculate throughput when executing in parallel across 64 banks of 4 modules. Latency of our method represents the time required from the execution of the first DRAM command to the completion of the last DRAM command for a single kernel execution which comprises multiple POPCNT3 operations. As shown in Table 2, our proposed method consistently outperforms the GPU across all kernel sizes. Specifically, our method achieves approximately 348 times higher throughput for the smallest kernel size (7x65536) and still maintains about 27 times higher throughput for the largest kernel size (127x65536). This significant improvement in throughput demonstrates that our proposed method effectively leverages the bit-level parallelism of DRAM to accelerate bulk bitwise accumulation operations. The ability to process data at such high rates is particularly beneficial for large-scale MAC operations, potentially enabling much faster execution of machine learning algorithms and neural network computations.

Table 2: Comparison of throughput and latency for bulk bitwise accumulation using our method and GPU (A100) for different kernel sizes. Our evaluation covers four dimensions: $K = 7, 15, 31, 63$ and $N = 65536$. We chose these $K$ values to represent realistic accumulations within a subarray, with output bit counts of exactly 3, 4, 5, and 6, respectively. $N$ is equal to the number of columns in the DRAM.

| Kernel Size | Our Method | | GPU (A100) | |
|---|---|---|---|---|
| K x N | Throughput (TB/s) | (Latency (us)) | Throughput (TB/s) | (Latency (us)) |
| 7x65536 | 47.3 | (4.97) | 0.136 | (7.06) |
| 15x65536 | 37.0 | (13.6) | 0.352 | (7.36) |
| 31x65536 | 32.2 | (32.3) | 0.609 | (5.95) |
| 63x65536 | 30.0 | (70.8) | 0.834 | (6.09) |
| 127x65536 | 28.6 | (149) | 1.04 | (6.48) |

Due to the inherent error in PuCD operations, our proposed bulk bitwise accumulation also has error. Table 3 presents the normalized mean square error (NMSE) of our DRAM-based calculations compared to ideal computational results (details in Appendix B). As shown in Table 3, the error in our proposed method is three to four orders of magnitude smaller than the quantization error typically used in machine learning [Wei et al., 2024]. This implies that adopting our method for neural network inference would have a negligible impact on inference accuracy.

Table 3: NMSE error of our bulk bitwise accumulation for different kernel sizes.

| Kernel Size | NMSE error |
|---|---|
| 7x65536 | 9.1e-08 |
| 15x65536 | 1.1e-07 |
| 31x65536 | 2.9e-07 |
| 63x65536 | 6.9e-07 |
| 127x65536 | 1.5e-06 |

## 5 Conclusion

This paper has introduced an efficient method for bulk bitwise accumulation using PuCD. Our approach addresses the challenges of implementing accumulation operations in PuCD, which are crucial for machine learning tasks. We presented a novel implementation of the POPCNT3 operation optimized for commercial DRAM, serving as a fundamental building block for our scalable bitwise accumulation technique. Evaluation using DDR4 DRAM modules shows that our approach

consistently outperforms GPU-based methods across all tested kernel sizes, with throughput improvements ranging from 27 to 348 times, while maintaining high accuracy. These results highlight the effectiveness of leveraging bit-level parallelism in DRAM for accelerating bulk bitwise accumulation operations. This substantial increase in throughput is crucial for constructing large matrix multiplications, which form the cornerstone of many machine learning inference tasks. The ability to process vast amounts of data rapidly can lead to significant advancements in real-time predictions and efficient handling of large datasets. Future work could explore integrating this approach into broader machine learning frameworks and extending it to other types of operations. By enabling efficient PIM capabilities in standard DRAM, our work contributes to addressing the data transfer bottleneck in modern computing systems.

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

## A    Evaluation Setup

Table 4 provides a detailed overview of the hardware specifications used in our experiments. The core of our setup consists of a Xilinx Alveo U200 FPGA board, which serves as the platform for implementing our PuCD-based method. We paired this FPGA with four SK Hynix DDR4 DIMM modules, each with a density of 4GB and operating at 2400MT/s. For precise control over DRAM operations, we employed DRAM Bender [Olgun et al., 2023], an open-source memory controller implemented on the FPGA. DRAM Bender is crucial to our setup as it enables us to issue DRAM commands with arbitrary timings, a capability essential for implementing PuCD operations. We controlled DRAM Bender through a host CPU program, allowing us to generate and execute the specific command sequences required for our bulk bitwise accumulation method.

Figure 4 shows the setup consisting of the following key components: ① A Xilinx Alveo U200 FPGA board programmed with DRAM Bender. This board serves as the central processing unit for our PuCD operations. ② Four DDR4 DIMM modules, each containing 8 DRAM chips, are installed in the DIMM slots of our setup. This configuration results in a total of 32 DRAM chips available for our experiments, allowing us to explore the full potential of bank-level parallelism in our method. ③ A separate computer, responsible for generating DRAM commands and controlling the overall experiment flow. This host machine runs the software that interfaces with DRAM Bender to execute our bulk bitwise accumulation operations.

Our measurement methodology involved conducting 1000 independent trials for each kernel size and configuration to ensure statistical significance and account for DRAM behavior variability. We focused on two primary metrics: throughput, calculated as the amount of data processed per second when executing our method in parallel across all 64 banks of the 4 DRAM modules, and latency, defined as the time elapsed from the first DRAM command to the last for a single kernel execution.

To provide a meaningful performance comparison, we implemented a baseline version of our bulk bitwise accumulation method on an NVIDIA A100-PCIe-40GB GPU using CUDA. GPU throughput was calculated based on the execution time of 8192 parallel kernel operations.

Table 4: Specifications of the FPGA and DRAM module used in our evaluation.

| FPGA | DRAM Module | Memory Bandwidth (GB/s) |
| --- | --- | --- |
| Xilinx Alveo U200 | SK Hynix's DDR4 DIMM [1] | 76.8 |

[1] We used four DRAM modules HMA851U6CJR6N-UHN0 that have a density of 4GB and operates at 2400MT/s.

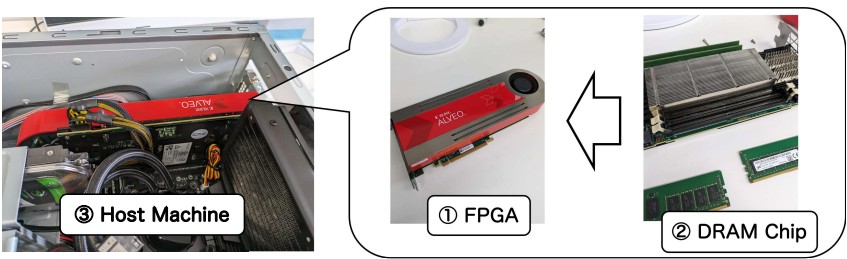

Figure 4: DRAM chips testing setup.

## B    NMSE Calculation

To quantify the accuracy of our bulk bitwise accumulation method, we employed the normalized mean square error (NMSE) metric NMSE is a statistical measure that quantifies the relative difference between predicted and actual values, normalized by the variance of the actual values. It is defined as:

$$\text{NMSE} = \frac{E[(Y - \hat{Y})^2]}{\text{Var}(Y)}$$

NMSE values provide a measure of our method's accuracy relative to the inherent variability in the accumulation task. Lower NMSE values indicate better performance, with an NMSE of 0 representing perfect prediction.

For this evaluation, we generated input bits randomly with a 50% probability for each bit state. Our NMSE calculation process began with performing an ideal accumulation using a high-precision software implementation for each set of randomly generated inputs. This provided our "actual" values (Y). We then performed the accumulation using our proposed PuCD-based method, obtaining our "predicted" values ($\hat{Y}$). For each column, we calculated the squared difference between the ideal and PuCD-based results: $(Y - \hat{Y})^2$. We then computed the mean of these squared differences across all columns, giving us $E[(Y - \hat{Y})^2]$. Next, we calculated the variance of the ideal accumulation results across all columns: $\text{Var}(Y)$. Finally, we divided the mean squared error by the variance to obtain the NMSE.

