# OpenReview forum: "Bulk Bitwise Accumulation in Commercial DRAM"
_NeurIPS.cc/2024/Workshop/MLNCP — MLNCP Oral_

### Official Review · Reviewer_p4Wa · 2024-10-01
**Impressing contribution with a great potential**

**Rating:** 10
**Confidence:** 4

**Review:**

The authors propose to use Processing-using-Commercial-DRAM (PuCD) to implement the POPCOUNT operation inside standard DRAM with a very high level of parallelism.

The authors first propose to compute the POPCNT3 operation, which is similar to a full adder in logic circuits, and produce a 2-bit result (one MSB and one LSB). The MSB is computed as MAX3(a,b,c), with a,b, and c as the inputs. Then, the LSB is computed as MAX5(!MSB,!MSB, a,b,c).

So simple and so powerful. This is one of the most significant contributions I have seen in five years!

For the second step, the authors propose decomposing the generic POPCOUNT operation into a set of POPCNT3 operations and reporting the timing for POPCNT7, POPCNT15, POPCNT31, POPCNT63, and POPCNT127. The concept is validated on an FPGA board, leading to throughputs up to 47TB/s compared to 0.136TB/s for an A100 GPU, POPCNT7 operation.

I have only one comment/suggestion: a small extra work is needed to fully implement a binary neural network where XNOR gates replace multiplications, and accumulations are POPCOUNT. Would you try to do it?

---

### Official Review · Reviewer_Zjid · 2024-10-03
**New use of MAJX to achieve population-count-of-3 and efficient scaling for bulk bitwise accumulation.**

**Rating:** 8
**Confidence:** 4

**Review:**

The authors use majx in a novel way to create a building block, popcount3, which can then be used to perform bitwise accumulation within a DRAM subarray efficiently and with fewer errors than using traditional logic operations. They note that they are able to improve throughput by 27-348x over the A100, which is quite impressive. This paper presents a simple and clever way to tackle the challenging problem of processing-in-memory accumulation. The paper is well written, and the figures help illustrate the process of the building blocks and extension. Overall, the work is clear for readers, and the value of their proposed approach is evident.


A few minor questions:
- I think the paper is good as is. Just curious--how important is it that the accumulation can only be done in a subarray, especially in the context of machine learning and deep learning? Can your ideas be used between subarrays for larger accumulation?
- It's nice that the authors show latency vs the kernel size, but it would be helpful to the reader if you could put these numbers in more familiar terms for those not in the field. Like what is the latency of adding two 32-bit precision numbers, or 16-bit numbers etc. using your method?

But overall, nice, clever paper.

---

### Decision · Program_Chairs · 2024-10-10

Accept (Oral)